# Diversity and Characterization of Resistance to Pyraclostrobin in *Colletotrichum* spp. from Strawberry

Shuodan Hu [1,†], Shuhan Zhang [1,†], Wenfei Xiao [2], Yahui Liu [1], Hong Yu [2] and Chuanqing Zhang [1,*]

[1]   College of Advanced Agricultural Sciences, Zhejiang A&F University, Lin'an, Hangzhou 311300, China; sdhu@stu.zafu.edu.cn (S.H.); 22316238@zju.edu.cn (S.Z.); lyhui0906@zafu.edu.cn (Y.L.)
[2]   Research Institute for the Agriculture Science of Hangzhou, Hangzhou 310013, China; xiao_wenfei@126.com (W.X.); yuh5060@gmail.com (H.Y.)
[*]   Correspondence: cqzhang@zafu.edu.cn
[†]   These authors contributed equally to this work.

**Abstract:** Strawberry crown rot poses a significant menace to strawberry production during the seedling stage, and the main pathogen is *Colletotrichum* spp. Pyraclostrobin is one of the main fungicides that have been registered to control anthracnose caused by *Colletotrichum* spp. The diversity of pathogens and the risk of fungicide resistance may change from year to year. In order to explore the diversity of pathogens causing crown rot and evaluate the resistance risk of pathogens to pyraclostrobin in different years, crown rot samples were collected in Jiande, Zhejiang Province in 2019 and 2021, and the pathogens were identified. Based on the morphological identification and phylogenetic analysis based on *ACT*, *CAL*, *CHS*, *GAPDH*, and ITS, all 55 strains were identified as *C. gloeosporioides* species complexes, including 23 *C. siamense* isolates and 2 *C. fructicola* isolates in 2019, and all isolates were identified as *C. siamense* in 2021. *C. siamense* was the dominant pathogen of strawberry crown rot in 2019 and 2021. The resistance frequencies of the isolates collected in 2019 and 2021 to pyraclostrobin were 69.57% and 100%, respectively. In general, compared to that in 2019, the resistance frequencies of the pathogen to pyraclostrobin increased in 2021. In terms of fitness, there was no significant difference between resistant strains and sensitive strains in the mycelium growth rate, sporulation and spore germination rate. In addition, the resistant mutants exhibited positive cross-resistance to kresoxim-methyl and azoxystrobin. A sequential analysis of cytochrome b gene showed that *C. siamense* resistance to pyraclostrobin is linked to the G143A point mutation. Our study indicated that the risk of resistance a fungicide gradually increases with the increase in use years, and in order to reduce the emergence and spread of resistant populations, we should choose fungicides of different mechanisms of action for rotation to reduce the risk of resistance development.

**Keywords:** strawberry; *C. siamense*; pyraclostrobin; resistance



## 1. Introduction

Strawberry is widely cultivated for its good economic benefits and high nutritional value [1]. With the increase in strawberry yield, fungal diseases have emerged as the main factor affecting the yield and quality of strawberry [2]. Strawberry crown rot is a major disease in most strawberry-growing areas, which can lead to serious economic losses. The crown is damaged, causing the above-ground part to be unable to absorb water and nutrients and eventual wilt [3]. Previous studies have shown that the primary pathogen of crown rot in Zhejiang Province is *Colletotrichum siamense*, followed by *C. fructicola* [4]. The diversity of this pathogen may change due to host immunity and environmental conditions [5]. It is not clear whether or not the dominant pathogens of crown rot have changed in recent years in strawberry-producing areas of Zhejiang Province. *Colletotrichum* spp. can be identified via their morphological characteristics such as colonies and spores, but these are easily affected by environmental factors [6]. Currently, numerous significant

pathogens are identified via multilocus phylogenetic analysis combined with morphological characteristics [7].

To effectively manage *Colletotrichum* spp., we are required to consider various control measures. In terms of chemical control, quinone outside inhibitors (QoIs) are one of the fungicides often used to control *Colletotrichum* spp. [8,9]. The mechanism of action of QoI fungicides involves binding to the ubiquinol-oxidizing (Qo) site, disrupting electron transfer on cytochrome b (*Cyt b*) and inhibiting ATP production [10]. QoIs fungicides have broad-spectrum activity against fungal species. Pyraclostrobin, a prominent QoI fungicide, is utilized for controlling numerous fungal diseases, since with their widespread applications, resistance has emerged among several plant pathogens [11,12]. Resistance to QoI fungicides may occur because of point mutations in the *Cyt b* gene. It is reported that amino acid mutations at G143A, F129L and G137R confer varying degrees of QoI resistance in *C. acutatum* [13]. G143A mutations are the most common in resistant strains and confer high levels of resistance. The mutations F129L and G137R are linked to moderate levels of resistance [14,15]. Forcelini BB et al. collected strawberry samples from 1994 to 2014 to assess the resistance of *C. acutatum* to QoIs. The results showed that with the continuous use of fungicides, the resistance frequency of *C. acutatum* to QoI fungicides increased year by year [13]. However, studies have also shown that the frequency of resistance gradually decreases with the use of fungicides [16]. Pyraclostrobin is the main QoI fungicide used to control strawberry crown rot. The resistance level of strawberry crown rot pathogens may change with the continuous use of pyraclostrobin.

Resistant mutations may impose fitness costs in the absence of fungicide selection pressure. Bauske et al. assessed the inherent fitness of pathogen mutants by measuring parameters such as the mycelium growth rate, spore production, and spore germination rate. The results indicated no significant difference in mycelium growth rate and spore germination rate between resistant and sensitive strains [17]. Similar results were obtained in *Magnaporthe grisea* mutants [18]. The fitness of G143A mutants of *M. grisea* was not affected and their resistance remained steady. Veloukas et al. reported that the biological characteristics of *Botrytis cinerea* with the G143A mutation were similar [19]. In addition, mutants resistant to pyraclostrobin are generally cross-resistant to fungicides of the same mechanism of action [20,21]. However, little is known about the fitness, competitiveness and cross-resistance of strains of *Colletotrichum* spp. that are resistant to pyraclostrobin.

The objectives of this study were to (i) define the dominant pathogen of strawberry crown rot in different years, (ii) determine the resistance frequency of *C. siamense* to pyraclostrobin in different years, and (iii) assess the fitness of resistant and sensitive strains without fungicide selection pressure.

## 2. Materials and Methods

### 2.1. Isolation of Colletotrichum spp.

Strawberry crown rot samples with brown vascular bundles and plant wilting were collected from the same strawberry production area in Jiande, Hangzhou Province (27°02′ to 31°11′ N; 118°01′ to 123°10′ E), in 2019 and 2021. The pathogen was isolated from strawberry crown via the tissue separation method [22]. The soil on the surface of the diseased tissues was cleaned with double-distilled water (ddH$_2$O), followed by being immersed for 30 s in 75% alcohol, being soaked for 2 min in 3% sodium hypochlorite, and being subsequently triple-rinsed with ddH$_2$O. The cleaned tissues were cut into $5 \times 5$ mm pieces using a sterile scalpel and placed onto potato dextrose agar (PDA) medium containing 100 mg/L of streptomycin sulfate. Cultures were incubated for 2–4 days at 28 °C, and then we transferred the mycelia to a fresh PDA plate [23]. The single-spore isolates were conserved on PDA at 4 °C.

### 2.2. Pathogenicity Assay

Pathogenicity tests were performed to determine whether or not the isolates were pathogenic. Five mycelia pieces (5 mm) were excised from colony's edge, transferred

to a 100 mL potato dextrose (PD) medium and oscillated at 175 r/min and 28 °C to induce conidial production [24]. The concentration of spore suspension was adjusted to $10^6$ conidia/mL. Six-week-old healthy strawberry plants were adopted for inoculation. An amount of 5 mL of the spore suspension was sprayed onto the strawberry crowns. The negative group was inoculated with ddH$_2$O, with each isolate inoculating five plants. The plants were returned to a greenhouse at 28 °C.

### 2.3. Morphological Characterization

The isolates stored at a 4 °C were activated on a PDA plate incubating for 3 days, and then transferred onto a new PDA plate. After incubating for 7 days, the diameter of the colony was measured, the growth rate was calculated, and colony morphology was observed and recorded [25]; the experiment was conducted in three replications. The mycelium was scraped from the surface of the PDA culture medium using a glass slide and rinses with ddH$_2$O. The mycelium was filtered through three layers of filter paper to obtain a spore suspension. The spore morphology was observed and the spore size was measured under a Scope A1 optical microscope (Nikon Eclipse 80i or Nikon SMZ25, Tokyo, Japan); fifty spores were measured for each isolate. An observation of appressorium was carried out in accordance with Emmett et al. [26], and the concentration of the spore suspension was set at $10^5$ conidia/mL; the 10 μL spore suspension was added to the center of the hydrophobic membrane, and cultured at 28 °C for 14 h. The cover glass was pressed on the hydrophobic membrane, appressorium morphology was observed and the size was measured under the Scope A1 optical microscope (Nikon Eclipse 80i or Nikon SMZ25, Tokyo, Japan). Each isolate was measured, resulting in fifty appressoriums.

### 2.4. Molecular Identification and Phylogenetic Analysis

Genomic DNA from each isolate was extracted using a fungi genomic DNA rapid extraction kit (B518229-0100; Sangon Biotech, Shanghai, China). The calmodulin (*CAL*), glyceraldehyde 3-phosphate dehydrogenase (*GAPDH*), chitin synthase (*CHS*), actin (*ACT*) and internal transcribed spacer (ITS) genes were amplified using the primer pairs CL1C/CL2C [27], GDF/GDR [28], CHS-79F/CHS-354R [29], ACT-512F/ACT-783R [30] and ITS-1F/ITS4 [31]. Phylogenetic analysis was carried out following previously described methods [32]. The gene numbers of standard strains utilized in this study are provided in Supplemental Table S1. All sequences were edited using MEGA 7.0 [33]. The clipped sequences were combined in Sequence Martix 1.8. Modeltest 3.7.win, Mrmodeltest2.3 and Win paup4b10-console, as implemented in MrMTgui v1.0, and were used to estimate the best nucleotide substitution model [34]. Bayesian inference (BI) phylogenies were constructed using Mr. Bayes v.3.1.2 [35]. Five simultaneous Markov chains were run for 300,000 generations each. Phylogenetic trees were visualized using Figtree v1.4.4 [36].

### 2.5. Resistance Detection

The molecular identification results showed that *C. siamense* was the dominant strain of crown rot, so the resistance level of 53 *C. siamense* to pyraclostrobin was measured using the differential measurement method. All isolates were cultured on PDA medium, and then transferred onto a new PDA plate. The mycelia plugs were prepared from the edge of the colony and selected and inoculated on the PDA medium plate containing different concentrations of pyraclostrobin. Using 0 μg/mL and 10 μg/mL as the differential concentrations [37], each concentration was inoculated with three PDA plates. After culture for 3 days at 28 °C, sensitive strains were defined as those able to grow at 0 μg/mL and unable to grow at 10 μg/mL, resistant strains were defined as those able to grow at 10 μg/mL. The frequency of resistance was calculated in accordance with the following formula [37]:

Resistance frequency (%) = (number of resistant strains/numbers of total strains) × 100

### 2.6. Sensitivity Level Determination

Five pyraclostrobin-resistant and five pyraclostrobin-sensitive strains were selected to determine the sensitivity of *C. siamense* to pyraclostrobin. The tested strains stored in freeze storage tubes were inoculated on the PDA medium and then transferred onto a new PDA plate. Mycelial plugs with a diameter of 5 mm were removed from the edge of the colony and inoculated onto PDA plates containing different concentrations of pyraclostrobin. Sensitive strains were inoculated in PDA medium containing 0.5, 1, 2, 4 and 8 μg/mL of pyraclostrobin, and resistant strains were inoculated in PDA medium containing 2, 4, 8, 16 and 32 μg/mL of pyraclostrobin. The PDA plate without pyraclostrobin was used as the control, and three PDA plates were measured for each concentration. After culture at 28 °C for 7 days, the cross-crossing method was used to measure the colony diameter, which was was measured horizontally and vertically, and calculated the inhibition rate of mycelia [38]. The virulence regression equation, effective inhibitory medium concentration ($EC_{50}$ value) and correlation coefficient were obtained by regressing the percent relative growth against the log10 values of the fungicide concentrations using SPSS 22.0 software.

### 2.7. Fitness Determination of Sensitive and Resistant Strains

In terms of mycelial growth, the strains stored at 4 °C were inoculated on a PDA plate, cultured at 28 °C for 3 days and transferred onto a new PDA plate. After incubation for 7 days, the colony diameter of each strains was measured in two approximately perpendicular directions; the mycelia growth rate was calculated. Three plates were measured for each strain.

In terms of sporulation, five mycelia plugs with a diameter of 5 mm were collected from the edge of the colony, transferred onto a 100 mL liquid potato dextrose (PD) medium. and shaken at 175 rpm and 28 °C to induce conidia production [24]. After 5 days, the mycelium was filtered with three layers of filter paper to obtain the spore suspension. Briefly, 10 μL of the spore suspension was pipetted onto a hemocytometer and spore production was counted under an A1 microscope. Three fields were observed for each strain.

In terms of the spore germination rate, the spore suspension concentration was adjusted to $10^5$ conidia/mL. Briefly, 20 μL of the spore suspension was taken and applied on water agar medium, cultured at 28 °C. A spore was considered germinated if the germ tube had reached at least half the length of the conidium. When the spore germination rate of the control group reached more than 90%, the data were counted [25]. The number of germinated spores was observed under the Scope A1 optical microscope (Nikon Eclipse 80i or Nikon SMZ25, Tokyo, Japan). We observed 5 fields for each strain, with more than 100 spores in each field.

### 2.8. Cross-Resistance

Kresoxim-methyl and azoxystrobin was used to assay the cross-resistance to pyraclostrobin [37]. Mycelia plugs with a diameter of 5 mm were taken from the edge of the colony and inoculated on the PDA plates with kresoxim-methyl and azoxystrobin in different concentrations. Sensitive strains were inoculated in PDA medium containing 0.5, 1, 2, 4 and 8 μg/mL of kresoxim-methyl/azoxystrobin, and resistant strains were inoculated in PDA medium containing 2, 4, 8, 16 and 32 μg/mL of kresoxim-methyl/azoxystrobin. After incubation for 7 days at 28 °C, the diameters of the colonies were measured via the cross-crossing method and inhibition rates were calculated to compute the $EC_{50}$ values. Each concentration was inoculated with three PDA plates.

### 2.9. Amplification and Sequencing of CsCyt b

Genomic DNA from both resistant and sensitive strains was extracted using the CTAB methods with minor modifications. The mycelium was scraped into 1.5 mL tube, quartz sand and 300 μL of CTAB extract were added, and the mixture was ground for 120 s at 85 Hz. Then, 400 μL CTAB extract was added and the mixture was shaken for 15 s. An amount of 700 μL of trichloromethane was added; the mixture was shaken for 15 s and

centrifuged at 12,000 rpm for 10 min. Next, 500 μL of the supernatant was removed and transferred into a new 1.5 mL tube; 500 mL of isopropyl alcohol was added, and the mixture was centrifuged at 12,000 rpm for 10 min. The supernatant was discarded; to it was added 500 mL of 70% ethanol, and then it was centrifuged at 12,000 rpm for 5 min. Finally, the supernatant was discarded, After dried, 50 μL of ddH$_2$O was added, and the mixture was stored at 4 °C. According to the *CsCyt b* gene sequence retrieved from NCBI, PCR primers (F: 5′-TCTGCTTTCTTCTTCTTAGTTTA-3′ and R: 5′GGGATAGCACTTATAAGGTTAGT-3′) were designed to amplify the fragment containing the complete coding region of *CsCyt b*. The PCR reaction system was 50 μL, the upstream and downstream primers were 2 μL each, the template was 1 μL, the 2× mix TaqDNA Mix was 25 μL, and ddH$_2$O was added to 50 μL. The reaction procedure was as follows: 95 °C for 3 min, followed by 35 cycles of 95 °C for 15 s, 56 °C for 15 s, 72 °C for 2 min, and finally 72 °C for 10 min. The PCR products were verified on 1.0% agarose gel at 254 nm (UV) and further sequenced by Tsingke Biotechnology Co., Ltd. (Beijing, China). Using Snapgene v5.2.4, the sequence was aligned and analyzed. The base sequence was compared and translated into amino acids after removing the introns, and the mutation of amino acids was analyzed.

## 3. Results

### 3.1. Isolation and Identification of Colletotrichum spp.

In 2019 and 2021, in total, 25 and 30 isolates of *Colletotrichum* spp. were isolated, respectively. In accordance with Koch's postulates, all isolates were demonstrated to be pathogenic. According to the morphological and cultural characterizations, all 55 isolates belonged to *C. siamense* and *C. fructicola*, including 23 *C. siamense* isolates and 2 *C. fructicola* isolates in 2019, while all isolates were identified as *C. siamense* in 2021. *C. siamense* colonies are grayish white on the front side and grayish on the back side. *C. fructicola* colonies have grayish white edges and a celadon center (Figure 1). No significant differences were observed in terms of mycelium growth rate and sporulation between the two *Colletotrichum* species, and the morphology and size of the spores and appressorium were similar (Table 1). *ACT*, *CAL*, *CHS*, *GADPH*, and ITS datasets were utilized for constructing a phylogenetic tree. All identified isolates were categorized within the *C. gloeosporioides* species complex, with 53 isolates belonging to *C. siamense* and 2 isolates belonging to *C. fructicola* (Figure 2). Supplementary Table S1 provides the GenBank numbers for all isolates.

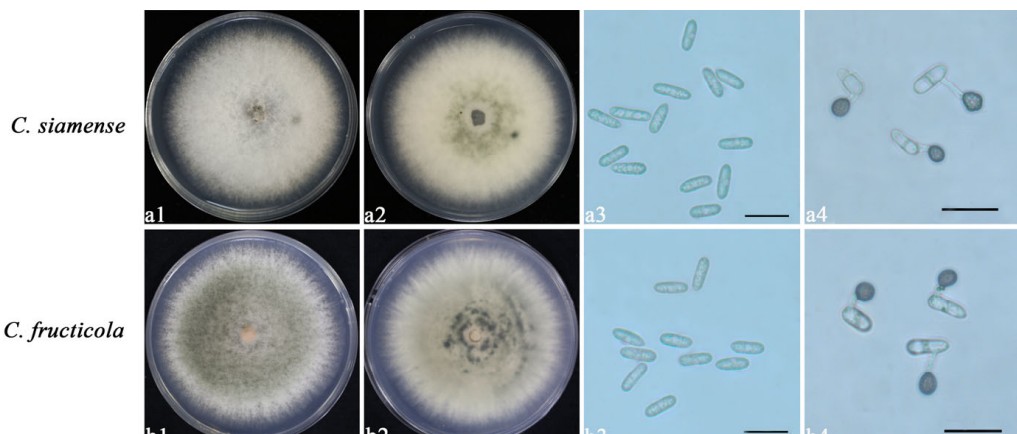

**Figure 1.** Colony morphology of of *Colletotrichum siamense* and *C. fructicola* on PDA (**a1,a2,b1,b2**), conidia (**a3,b3**), and appressoria (**a4,b4**); scale bar = 20 μm in (**a3,b3,a4,b4**).

**Table 1.** Size of spore, appresoria, hyphae growth rates, and sporulation of *Colletotrichum siamense* and *C. fructicola*.

| Species | Strain Number | Conidia [y] | | Appressoria [y] | | Growth Rate (mm/Day) [y] | Sporulation (×10⁶) [y] |
|---|---|---|---|---|---|---|---|
| | | Length (μm) | Width (μm) | Length (μm) | Width (μm) | | |
| *C. siamense* | JD-HY-20 | 15.26 ± 0.80 a | 5.55 ± 0.22 a | 6.58 ± 0.33 a | 5.92 ± 0.57 a | 10.91 ± 0.11 a | 15.25 ± 2.80 a |
| | H35 | 15.60 ± 0.36 a | 5.83 ± 0.19 a | 6.79 ± 0.15 a | 6.13 ± 0.36 a | 11.51 ± 0.65 a | 15.6 ± 3.36 a |
| *C. fructicola* | JD-ZJ-5 | 16.83 ± 0.14 a | 5.86 ± 0.14 a | 6.19 ± 0.15 a | 6.59 ± 0.32 a | 12.58 ± 0.22 a | 16.06 ± 4.14 a |
| | JD-ZJ-9 | 16.64 ± 0.23 a | 6.74 ± 0.15 a | 6.70 ± 0.42 a | 6.37 ± 0.23 a | 12.67 ± 0.12 a | 16.83 ± 3.23 a |

[y] Data are the mean ± standard error. The same letters represent the data being not statistically different ($p > 0.05$) according to the least significant different (LSD) test.

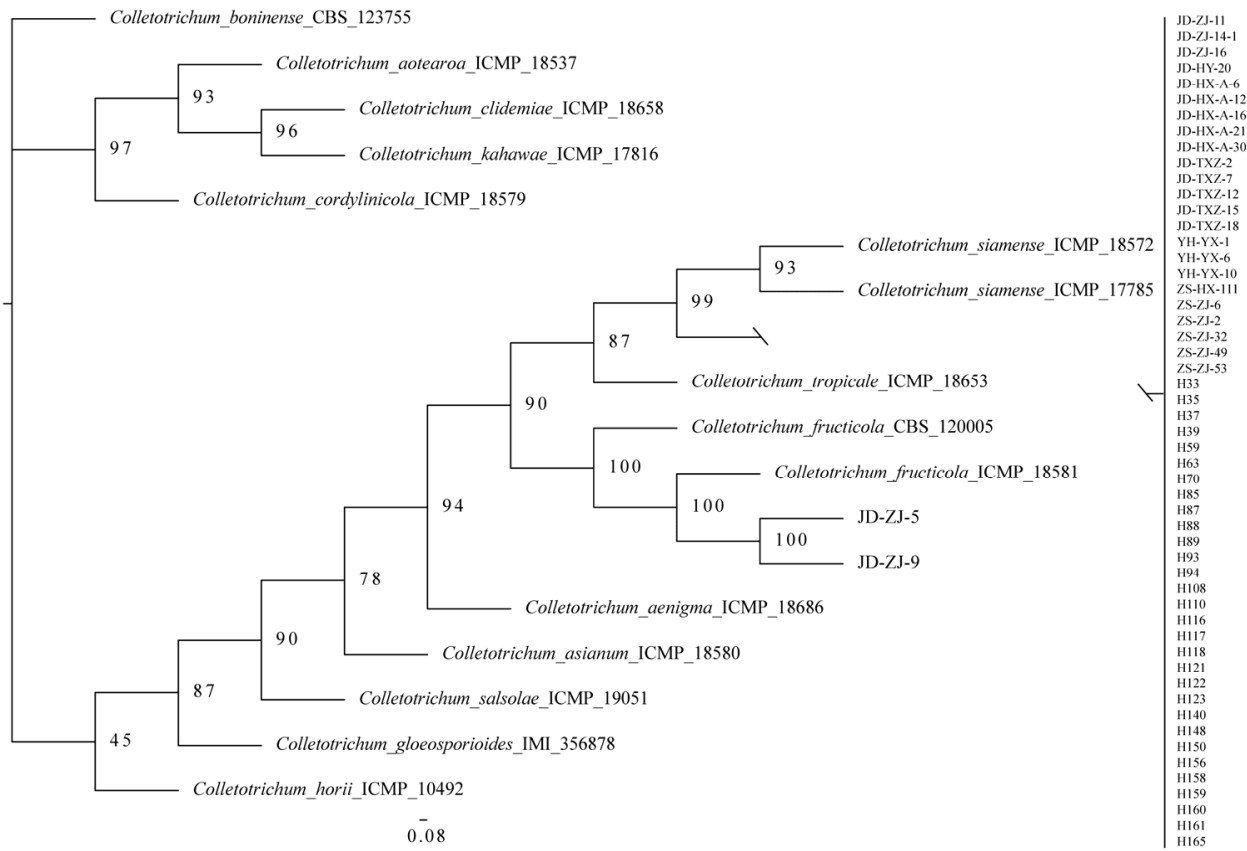

**Figure 2.** The *Colletotrichum gloeosporioides* species complex was determined as the identity of the phylogenetic tree using Bayesian inference. The sequencing analysis included the *ACT*, *CAL*, *CHS*, *GAPDH*, and ITS genes DNA sequence. The scale bar shows 0.08 expected changes per site.

### 3.2. Frequency of Resistance of C. siamense to Pyraclostrobin

The primary pathogen responsible for strawberry crown rot was *C. siamense*. Therefore, the resistance level of *C. siamense* to pyraclostrobin was assessed. Briefly, all 46 strains could grow on the PDA plates containing 10 μg/mL of pyraclostrobin, and were considered resistant strains; the resistance frequency was 86.79%. In 2019, 7 of the 23 *C. siamense* strains could not grow on the PDA plates containing 10 μg/mL of pyraclostrobin, and the other 16 strains grew on the PDA plates containing 10 μg/mL of pyraclostrobin and were resistant strains; the resistance frequency was 69.57% (Figure 3). In 2021, 30 strains were grown on PDA plates containing 10 μg/mL of pyraclostrobin with a resistance frequency of 100% (Figure 3).

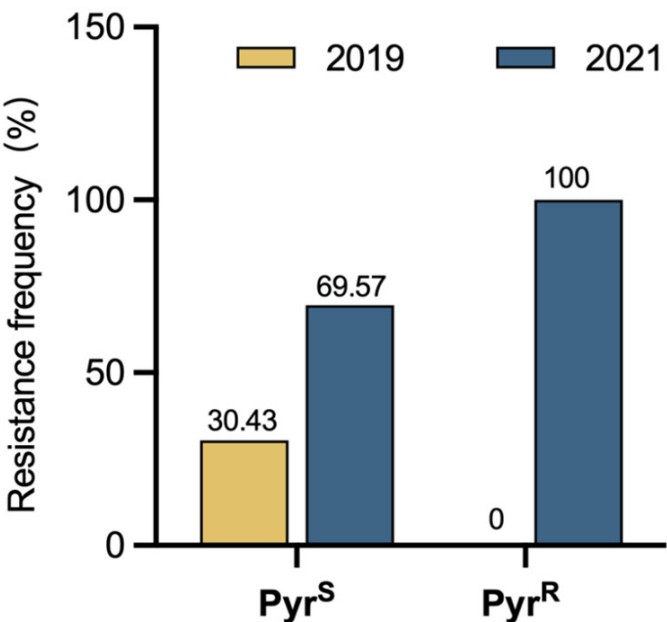

**Figure 3.** Frequency of resistance to pyraclostrobin in *Colletotrichum siamensis* in 2019 and 2021. Pyr$^R$ represents pyraclostrobin-resistant strain, and Pyr$^S$ represents pyraclostrobin-sensitive strain.

*3.3. Sensitivity of C. siamense to Pyraclostrobin*

To confirm the sensitivity of strains resistant and sensitive to pyraclostrobin, we calculated their EC$_{50}$ values of resistant and sensitive strains. The EC$_{50}$ values for sensitive strains varied from 1.192 to 2.068 μg/mL, and the average EC$_{50}$ values were 1.586 μg/mL (Table 2). The EC$_{50}$ values of resistant strains varied from 18.159 to 23.797 μg/mL, with an average EC$_{50}$ value of 21.158 μg/mL. The mean resistance level was 13.3.

**Table 2.** The EC$_{50}$ values of strains resistant and sensitive to pyraclostrobin.

| Isolate | Origin | EC$_{50}$ (μg/mL) |
|---|---|---|
| JD-HX-A-12-1 (Pyr$^S$) | Crown of strawberry | 1.544 |
| ZS-ZJ-111-1 (Pyr$^S$) | Crown of strawberry | 1.695 |
| JD-ZJ-11 (Pyr$^S$) | Crown of strawberry | 1.192 |
| JD-ZJ-14-2 (Pyr$^S$) | Crown of strawberry | 1.431 |
| JD-HY-20 (Pyr$^S$) | Crown of strawberry | 2.068 |
| H121 (Pyr$^R$) | Crown of strawberry | 20.974 |
| H211 (Pyr$^R$) | Crown of strawberry | 23.797 |
| H63 (Pyr$^R$) | Crown of strawberry | 18.159 |
| H116 (Pyr$^R$) | Crown of strawberry | 21.740 |
| H85 (Pyr$^R$) | Crown of strawberry | 21.120 |

Note: Pyr$^R$ represents pyraclostrobin-resistant strain, and Pyr$^S$ represents pyraclostrobin-sensitive strain.

*3.4. Fitness Comparison between Resistant and Sensitive Strains*

The mycelial growth rate, sporulation quantity and spore germination rate of the resistant and sensitive strains were measured. The results indicated no significant difference in the fitness between the resistant and sensitive strains (Figure 4). The growth rate of the resistant and sensitive strains was (11.16 ± 0.09) mm/d and (11.27 ± 0.06) mm/d, respectively. The sporulation quantities were (1.46 ± 0.11) × 10$^6$ conidia/mL and (1.34 ± 0.14) × 10$^6$ conidia/mL, respectively. The respective germination rates were (82.15 ± 1.19)% and (85.54 ± 2.31)%, respectively.

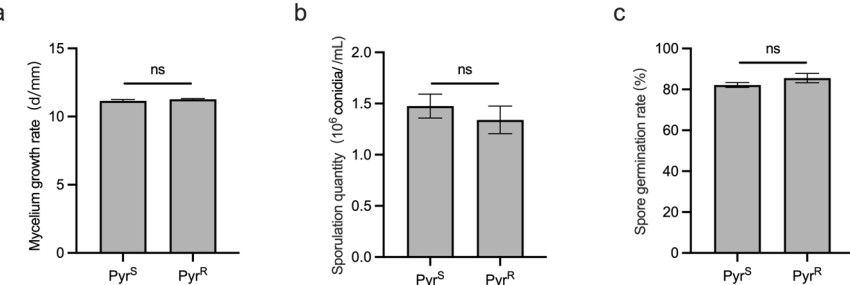

**Figure 4.** Comparison of mycelium growth rate (**a**), sporulation quantity (**b**) and spore germination rate (**c**) between pyraclostrobin-resistant and -sensitive strains of *Colletotrichum siamense*. $Pyr^R$ represents the pyraclostrobin-resistant strain, $Pyr^S$ represents the pyraclostrobin-sensitive strain, and "ns" means no significant difference.

### 3.5. Cross-Resistance of Resistant Mutants

Assessing cross-resistance is crucial for choosing fungicides to control plant pathogens that have developed resistance to specific fungicides. A total of 6 strains were tested for cross-resistance. Consequently, we examined the cross-resistance of pyraclostrobin-resistant strains to kresoxim-methyl and azoxystrobin. The findings demonstrated a significant positive cross-resistance between pyraclostrobin and kresoxim-methyl, azoxystrobin ($\rho > 0.8$) (Figure 5).

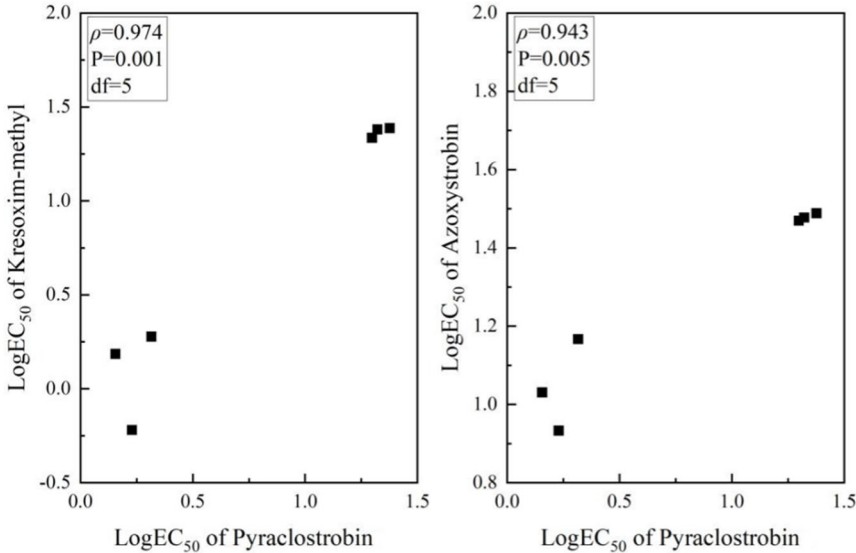

**Figure 5.** Cross−resistance between pyraclostrobin, and kresoxim−methyl and azoxystrobin. $p < 0.05$ indicates there is a significant difference. $\rho > 0.8$ indicates that there is cross-resistance. The black squares represent different strains.

### 3.6. Sequence Analysis of CsCyt b

In order to analyze whether or not the *cytb* gene of the pyraclostrobin-resistant strain had been mutated, we used primers LT-CytbF3 and LT-CytbR3 to amplify the *Cyt b* gene fragment of the pyraclostrobin-sensitive and -resistant strains, and a PCR product with a length of 2118 bp was obtained. The amino acid sequence was compared using Snapgene, and the amino acid at the 143rd position of the resistant strain had changed from G (Gly) to Ala (A) (Figure 6).

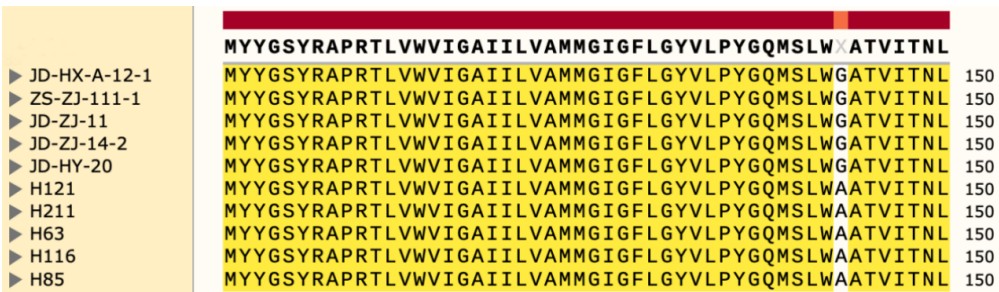

**Figure 6.** Comparison of *Cyt b* gene sequences between pyraclostrobin-resistant and -sensitive strains of *Colletotrichum siamense.* JD-HX-A-12-1, ZS-ZJ-111-1, JD-ZS-11, JD-ZJ-14-2, and JD-HY-20 are pyraclostrobin-sensitive strains; H121, H211, H63, H116, and H85 are pyraclostrobin-resistant strains.

## 4. Discussion

Through morphological identification and phylogenetic analysis, the pathogens of strawberry crown rot in 2019 and 2021 were identified as *C. gloeosporioides* species complexes. Furthermore, 23 *C. siamense* and 2 *C. fructicola* species were isolated in 2019, and all 30 isolates were *C. siamense* in 2021. *C. siamense* was the advantage pathogen of crown rot in 2019 and 2021, aligning with findings from previous studies. *C. siamense* was still the dominant pathogen of crown rot in different years [39]. There was no significant difference between *C. siamense* and *C. fructicola* in the mycelium growth rate, sporulation quantity, and spore and appressorium size. Previous studies have shown that the appressoria and spores of *C. fructicola* were larger [39] than those in this study, probably due to the small sample size in this study, which was not representative.

The application of fungicides is a critical measure with which to control anthracnose disease [9]. QoI fungicides are extensively employed to control crown rot induced by *Colletotrichum* spp., due to their broad spectrum [40]. However, the excessive use of the same fungicide can lead to selection pressure and the emergence of resistance [41]. *C. siamense* is the predominant pathogen responsible for strawberry crown rot in Zhejiang Province; therefore, we studied the resistance risk of *C. siamense* to pyraclostrobin in different years. In general, the resistance frequency of *C. siamense* to pyraclostrobin was 86.79%, of which the resistance frequency in 2019 was 69. 57%. The resistance frequency reached 100% in 2021; the resistance frequency of *C. siamense* to pyrazinamide increased. It is indicated that *C. siamense* has developed extensive resistance to pyraclostrobin.

Mutations in the *Cyt b* gene have been associated with resistance for QoI fungicides. Resistance to QoIs is usually caused by F129L, G137R and G143A substitution in mitochondrial *Cyt b* genes [13]. The 143rd amino acid mutation of *Cyt b* causes a high level of resistance to pyraclostrobin [35]. F129L has also been reported in *C. cereale*, *Magnaporthe grisea* and other pathogenic fungi [42]. In our study, we amplified and sequenced the *Cyt b* gene fragments from both sensitive and resistant strains. Only the point mutation of G143A (from glycine to alanine) was detected in resistant strains, suggesting high levels of resistance. F129L and G137R loci mutations were not found in our study. All *C. siamense*-resistant isolates showed a high level of resistance to pyraclostrobin in the greenhouses.

Resistance to QoI fungicides linked to G143A mutations can affect the structure of the Qo site, and this may reduce the activity of the cytochrome bc1 complex, thereby differentially affecting the adaptability of pathogens [43,44]. No notable distinctions were observed in the growth rate, sporulation rate and spore germination rate between resistant and sensitive strains for *C. siamense* from strawberry. These results indicated that a high risk of further spread of resistant isolates in the fields. Previous studies had shown adaptive penalties for QoI-resistant isolates in *C. gloeosporioides* [45], while no significant differences were found between QoI resistance and sensitivity isolates in the *C. acutatum* species complex [46]. These outcomes highlighted that different *Colletotrichum* species had different phenotypes for adaptive punishment in resistant isolates. This further indicated that the species of pathogen should be clearly defined in the prevention and control of disease.

It is easy for cross-resistance to occur between fungicides with the same mechanisms of action. Cross-resistance analysis is of great significance for the rational selection of fungicides and the control of plant pathogens resistant to one or more fungicides. We further analyzed the cross-resistance between pyraclostrobin and kresoxim-methyl and azoxystrobin, and found that there was cross-resistance between these three types of QoI fungicides. This is also consistent with previous reports [47]. This study showed that *C. siamense* developed widespread high-level resistance to pyraclostrobin and that there was no significant difference in biological characteristics between pyraclostrobin-resistant and sensitive strains for *C. siamense*. It may spreaded widely in the field. In general, *C. siamense* had a high risk of resistance to pyraclostrobin. Fungicides with different action modes should be replaced to control strawberry crown rot. The results of this study had important guiding significance for the formulation of specific anthracnose control strategies.

## 5. Conclusions

*C. siamense* remains the primary pathogen of strawberry crown rot in Zhejiang Province. Strains resitsant to pyraclostrobin have been found in the field and they exhibit high-level resistance. Biological characteristics do not significantly differ between pyraclostrobin-resistant and -sensitive strains. Furthermore, pyraclostrobin exhibits positive cross-resistance along with two other QoI fungicides, indicating that such fungicides should not be further employed for the management of strawberry crown rot.

**Supplementary Materials:** The following supporting information can be downloaded at https://www.mdpi.com/article/10.3390/agronomy13112824/s1. Table S1: *Colletotrichum* spp. used in multi-gene analysis in this study.

**Author Contributions:** Conceptualization, H.Y. and C.Z.; methodology, Y.L., H.Y. and C.Z.; software, S.H. and W.X.; validation, H.Y. and C.Z.; formal analysis, Y.L., W.X. and S.H.; investigation, S.Z. and S.H.; writing—original draft preparation, S.Z. and S.H.; writing—review and editing, H.Y. and C.Z.; visualization, S.Z. and S.H.; supervision, H.Y. and C.Z. All authors have read and agreed to the published version of the manuscript.

**Funding:** This research was funded by Agriculture and Social Development Research Project of Hangzhou (202203A07), and the Joint-Extension Project of important Agriculture Technology in Zhejiang Province (2021XTTGSC02-4).

**Data Availability Statement:** Data are contained within the article.

**Conflicts of Interest:** The authors declare no conflict of interest.

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
