# Peer review of "Diversity and Characterization of Resistance to Pyraclostrobin in Colletotrichum spp. from Strawberry"

_agronomy, doi:10.3390/agronomy13112824_

Round 1
Reviewer 1 Report
Comments and Suggestions for Authors
I may suggest the authors the revision of the manuscript according to the following suggestions and comments.
Abstract:
Lines 15 and 16. Please revise the sentence or complete the idea, the text is not clear.
Lines 16. Please specify what it was isolated.
Line 20. The phrase “C. siamense is still the main pathogen of strawberry crown rot” is out of context, please use a connector or revise de text.
Line 20. “The resistance frequencies of the isolates” please include resistance to x?, in order to make the text clear.
Line 22. Please complete the text and include the necessary information to clarify the idea.
The abstract needs to be revised: there are incomplete ideas, punctuation marks are not properly used, the grammar of some sentences seems not correct. There are sentences without any connection.
Introduction:
Lines 39 and 40. Please revise this phrase, the text is not clear.
Lines 41. The article the is missed “the pathogen”.
Lines 45. Please revise this phrase, the text is not clear.
Lines 47 and 48. Please revise the text; there is redundancy of terms.
Lines 49. Please revise this phrase, the text is not clear.
Lines 60 to 62. Please explain better the idea, it seems a bit confusing.
Line 79. The authors must check the objective 1. Please revise the text.
Line 84. Colletotrichum spp. is not correctly written.
Line 87. Please revise this phrase, the text is not clear.
Lines 87 to 93. Please revise the text and include the information missed to make the paragraph clear. Please include the references.
Line 96. Please revise: ¿it was measured the mycelium diameter or the diameter of the colony?
Line 98. Please explain what “each isolate was repeated 3 times” means.
Line 98 to 107. Please revise the paragraph and presented it in a clear way.
Line 127. Please explain what “different mass concentrations” means.
Line 131. Please include the reference. Due to the fact that it is a simple equation, it could be explained as text.
Line 137. Please explain what “The tested strains were activated” means.
Line 140. Different concentrations of? Please make it clear.
Line 142. Please explain what “without additive” means; please clarify what “each concentration was repeated 3 times” means, maybe authors area talking about three Petri dishes?
Line 147 and 147. Please explained how “virulence regression equation, effective inhibitory medium concentration (EC50 146 value) and correlation coefficient” were obtained. Please include the references that support the analysis conducted. In this case specify the variables of the pathogen that were considered in the “virulence” analysis.
Line 157. Please define PD liquid medium.
Line 159. I consider the expression “spore liquid were absorbed” not precise. Please use suitable terms to describe common procedures at laboratory in the study of plant pathogens.
Line 163. Please specify when a spore was considered as germinated and add the reference.
Lines 149 and 165. Please revise this paragraph, the text is not clear.
Lines 169 and 179. Please complete the sentence “with different concentrations” of?
The first and second objectives of the study should be revised.
The rationale of the study is not clearly presented.
Results
Line 189. Colletotrichum spp. is not correctly written, please revise scientific names.
Line 191. Please explain “all isolates were pathogenic”. It was not mentioned before that pathogenicity tests were conducted. If the results are available, include them and also the corresponding methodology.
Line 204. In figure 1 please specify which image is Colletotrichum siamense and which one correspond to C. fructicola. Please indicate the size of the scale bar.
Table 1. Please include all the necessary information in the table´s caption to make it clear
Table 1. The strain numbers for C. siamense are missed, please include them.
Lines 229 and 230. Please revise the phrase.
Table 2. Please include the origin of each isolate presented in the table.
Figure 4. Please include if the presented results are from C. siamense.
Figure 5. Please include all the necessary information in the figure´s caption to make it clear.
Figure 6. Please include all the necessary information in the figure´s caption to make it clear
In general, the authors must strengthen this part of the manuscript. To expand the presentation of the obtained results is worth the effort.
Discussion
Line 270. The authors should explain the ideas here presented.
Line 277. The authors may consider another way to describe “broad-spectrum”; in my opinion, it is more a characteristic than an ability.
Line 280. Please revise this phrase. The study was conducted in 2019 and 2021, therefore the authors may discuss the results in this context.
Line 282. Please revise this part of the phrase: “and the resistance risk will increase year by year with the extension of the year of application”. No results or any further analysis about this important topic is presented in the manuscript. The authors should explore more in depth the term “resistance risk”. Additionally, it was also an objective of the study.
The obtained results should be discussed in depth. Authors are just mentioning the results and if they agree or disagree with previous results. The authors should strengthen this part of the manuscript. There is a lack of analysis of the obtained results.
Line 312. The contribution of the manuscript is important and confirms results obtained previously. Authors mentioned: “The results of this study have important guiding significance for the formulation of specific anthracnose control strategies”. Due to the fact that anthracnose is an important disease in many crops, authors should present clearly what they think is a contribution of the study in “the formulation of specific anthracnose control strategies”.
The conclusions of the study should be presented.
Author Response
Review Report (Reviewer 1)
I may suggest the authors the revision of the manuscript according to the following suggestions and comments.
Abstract:
Lines 15 and 16. Please revise the sentence or complete the idea, the text is not clear.
Response: Changed to “In order to explore the diversity of pathogen causing crown rot and evaluate the resistance risk of pathogen to pyraclostrobin in different years.”
Lines 16. Please specify what it was isolated.
Response: The manuscript has already described the specific isolates, so this sentence has been deleted.
Line 20. The phrase “C. siamense is still the main pathogen of strawberry crown rot” is out of context, please use a connector or revise de text.
Response: Ok, changed to C. siamense was main pathogen of strawberry crown rot in 2019 and 2021
Line 20. “The resistance frequencies of the isolates” please include resistance to x?, in order to make the text clear.
Response: Ok. It has been added. The resistance frequencies of the isolates collected in 2019 and 2021 to pyraclostrobin were 69.57% and 100%, respectively.
Line 22. Please complete the text and include the necessary information to clarify the idea.
Response: In general, compared to 2019, the resistance frequencies of the pathogen to pyraclostrobin increased in 2021.
The abstract needs to be revised: there are incomplete ideas, punctuation marks are not properly used, the grammar of some sentences seems not correct. There are sentences without any connection.
Response: I have revised the abstract according to your suggestions, including the content and grammar to make it clear.
Introduction:
Lines 39 and 40. Please revise this phrase, the text is not clear.
Response: This phrase has been modified to “Previous studies have shown that the main pathogen of strawberry crown rot in Zhejiang Province is C. siamense, followed by C. fructicola.”
Lines 41. The article the is missed “the pathogen”.
Response:I have added it.
Lines 45. Please revise this phrase, the text is not clear.
Response:This phrase has been modified to “At present, multilocus phylogenetic analysis combined with morphological characteristics is used to identify many important pathogens”.
Lines 47 and 48. Please revise the text; there is redundancy of terms.
Response:This phrase has been modified to “The prevention and control of Colletotrichum spp. require the consideration of various control measures. In terms of chemical control, quinone outside inhibitors (QoIs) is one of the fungicides often used to control Colletotrichum spp.“.
Lines 49. Please revise this phrase, the text is not clear.
Response:This phrase has been modified to “The mechanism of action of QoI fungicides involves binding to the ubiquinol oxidizing (Qo) site, disrupting electron transfer on cytochrome b (Cyt b) and inhibiting ATP production”.
Lines 60 to 62. Please explain better the idea, it seems a bit confusing.
Response:I have explained this in the manuscript. Forcelini BB et al. collected strawberry samples from 1994 to 2014 to determine the resistance level of C. acutatum to QoIs. The results showed that with the continuous use of fungicides, the resistance frequency of C. acutatum to QoI fungicides increased year by year.
Line 79. The authors must check the objective 1. Please revise the text.
Response:Ok.I have revised it.The objectives of this study were to (i) define the dominant pathogen of strawberry crown rot in different years.
Line 84. Colletotrichum spp. is not correctly written.
Response:By reviewing relevant literature, "Colletotrichum spp." is the correct.
Line 87. Please revise this phrase, the text is not clear.
Response:Ok.I have revised it."The pathogen was isolated from strawberry crown by tissue separation method".
Lines 87 to 93. Please revise the text and include the information missed to make the paragraph clear. Please include the references.
Response:Ok.I have revised it to make clear.
Line 96. Please revise: ¿it was measured the mycelium diameter or the diameter of the colony?
Response:Ok.I have revised it.“The diameter of the colony”
Line 98. Please explain what “each isolate was repeated 3 times” means.
Response:Each strain is inoculated with three PDA plates, which means three repetitions
Line 98 to 107. Please revise the paragraph and presented it in a clear way.
Response:I have revised this paragraph to make it more clear.
Line 127. Please explain what “different mass concentrations” means.
Response:It means different concentrations. I have revised it.
Line 131. Please include the reference. Due to the fact that it is a simple equation, it could be explained as text.
Response:The reference has been added.
Line 137. Please explain what “The tested strains were activated” means.
Response:It means “The tested strains stored in freeze-storage tubes were inoculated on the PDA medium”.I have revised it.
Line 140. Different concentrations of? Please make it clear.
Response:Ok, I have revised it
Line 142. Please explain what “without additive” means; please clarify what “each concentration was repeated 3 times” means, maybe authors area talking about three Petri dishes?
Response:“without additive” means PDA without adding fungicide.“each concentration was repeated 3 times” means three Petri dishes.I have revised it in the manuscript to make it clear.
Line 147 and 147. Please explained how “virulence regression equation, effective inhibitory medium concentration (EC50 146 value) and correlation coefficient” were obtained. Please include the references that support the analysis conducted. In this case specify the variables of the pathogen that were considered in the “virulence” analysis.
Response: Ok, I have explained it in the manuscript make to clear. “Using SPSS 22.0 software, virulence regression equation, effective inhibitory medium concentration (EC50 value) and correlation coefficient were obtained by regressing the percent relative growth against the log10 values of the fungicide concentrations”.
Line 157. Please define PD liquid medium.
Response:Ok,I have added it in the manuscript, it is potato dextrose (PD) liquid medium.
Line 159. I consider the expression “spore liquid were absorbed” not precise. Please use suitable terms to describe common procedures at laboratory in the study of plant pathogens.
Response:I understand what you mean and I have revised the sentence.Changed to “use pipette to take 10 μL of spore liquid into the hemocytometer”
Line 163. Please specify when a spore was considered as germinated and add the reference.
Response:When the spore germination rate of the control group reached more than 90%, the data were counted.the reference has been added.
Lines 149 and 165. Please revise this paragraph, the text is not clear.
Response:I have revised this paragraph to make it more clear
Lines 169 and 179. Please complete the sentence “with different concentrations” of?
Response: Ok,I have added it in the manuscript.
The first and second objectives of the study should be revised.
The rationale of the study is not clearly presented.
Response:I have revised and improved the introduction according to your suggestions.And the rationale of this study is clearly presented.
Results
Line 189. Colletotrichum spp. is not correctly written, please revise scientific names.
Response:By reviewing relevant literature, "Colletotrichum spp." is the correct.
Line 191. Please explain “all isolates were pathogenic”. It was not mentioned before that pathogenicity tests were conducted. If the results are available, include them and also the corresponding methodology.
Response: Ok, I've added the method of pathogenicity test in 2.2
Line 204. In figure 1 please specify which image is Colletotrichum siamense and which one correspond to C. fructicola. Please indicate the size of the scale bar.
Response:I have modified the figure 1 and the scale bar = 20 µm
Table 1. Please include all the necessary information in the table´s caption to make it clear
Response:I have added it.
Table 1. The strain numbers for C. siamense are missed, please include them.
Response:Sorry, this is my oversight, I have added it.
Lines 229 and 230. Please revise the phrase.
Response:I have revised it, “To confirm the sensitivity of resistant and sensitive strains to pyraclostrobin, the EC50 values of resistant and sensitive strains were calculated”.
Table 2. Please include the origin of each isolate presented in the table.
Response:I have indicated the origin of each isolate in the table.
Figure 4. Please include if the presented results are from C. siamense.
Response:Yes, I have added it
Figure 5. Please include all the necessary information in the figure´s caption to make it clear.
Response:I have added the necessary information to make it clear.
Figure 6. Please include all the necessary information in the figure´s caption to make it clear
Response:I have added the necessary information to make it clear.
In general, the authors must strengthen this part of the manuscript. To expand the presentation of the obtained results is worth the effort.
Response:I have strengthened this part of the manuscript according to your suggestion.Make the results part more meaningful.
Discussion
Line 270. The authors should explain the ideas here presented.
Response:I have explain the ideas in the manuscript.“C. siamense was main pathogen of strawberry crown rot in 2019 and 2021. This is consistent with previous research. C. siamense was still the dominant pathogen of crown rot in different years”.
Line 277. The authors may consider another way to describe “broad-spectrum”; in my opinion, it is more a characteristic than an ability.
Response:I agree with you. I've turned “ability” into “characteristic”.
Line 280. Please revise this phrase. The study was conducted in 2019 and 2021, therefore the authors may discuss the results in this context.
Response:I have revised it.
Line 282. Please revise this part of the phrase: “and the resistance risk will increase year by year with the extension of the year of application”. No results or any further analysis about this important topic is presented in the manuscript. The authors should explore more in depth the term “resistance risk”. Additionally, it was also an objective of the study.
The obtained results should be discussed in depth. Authors are just mentioning the results and if they agree or disagree with previous results. The authors should strengthen this part of the manuscript. There is a lack of analysis of the obtained results.
Response: I have revised this sentence, high resistance frequency does not necessarily indicate high resistance risk.I have explored more in depth the term “resistance risk in discussion and strengthened this part of the manuscript
Line 312. The contribution of the manuscript is important and confirms results obtained previously. Authors mentioned: “The results of this study have important guiding significance for the formulation of specific anthracnose control strategies”. Due to the fact that anthracnose is an important disease in many crops, authors should present clearly what they think is a contribution of the study in “the formulation of specific anthracnose control strategies”.
The conclusions of the study should be presented.
Response:Ok, I have presented this conclusion in the manuscript. In general, C. siamense has a high risk of resistance to pyraclostrobin. Fungicides with different action modes should be replaced to control strawberry crown rot.
Reviewer 2 Report
Comments and Suggestions for Authors
Dear Authors, the content of the manuscript is interesting because it gives useful information for diases control in strawberry. Tha experimental design is appropriate, anyway an extensive english editing is required and a more accurate description of methods and results is needed. The major observation are reported below but a complete rewriting of the manuscript could valorize your research
Abstract
Lines 11-30: English editing needs improvement
Introduction: lines 44-45. “Colletotrichum spp. can be identified by morphological characteristics such as colonies and spores, but it is variable”. What is variable?
Line 63 -65: “With the continuous use of pyraclostrobin, it is unclear whether the resistance level of strawberry crown rot pathogens has changed.” It is suggested to move this sentence to hypothesis description
Line 87: please, rephrase “The pathogens were isolated by tissue isolation”
Line 92: “The single-conidium isolates”- Did the authors mean “monosporic cultures”?
Line 96: why the authors report this timing “after incubating for 3 days” if mycelium diameter was measured after 7 days? Can the sentence be deleted?
Line 98: the experiment was done in three replication and not “each isolate was repeated 3 times”, I suppose
Line 89-99: rephrase.
Line 101: 50. Write in letters
Line 109-110: please, rephrase.
Line 123: “The main pathogens of strawberry crown rot were C. siamense” this sentence have to be moved in results section. The authors could write: “The isolate mostly represented on the basis of molecular identification…..”
Line 125: “The main pathogens of strawberry crown rot were C. siamense” this sentence is redundant.
Line 126: What did the authors mean with “edge”?
Line 129-130: please, rephrase
Section 2.4 and 2.5: I do not understand why the contents of these two section is separated. Did the authors do a pre-screening and then a more accurate evaluation?
Line 144: please, add more information about cross crossing method
Line 162: what did the authors mean with “coated”?
Line 167-168: please, rephrase
Line 178: describe extraction method
Line 191: describe in materials and methods how the pathogenicity tests were performed
Line 196-197: give more element to say that the two species have no substantial differences. Briefly explain table 1
Figure 2: describe in the text the results in figure 2
Line 258: this sentence is incorrect.
Line 262: describe better this analysis in the method and result section
Introduction and discussion: as a general suggestion, the introduction is too long and several citation could be moved to discussion to better discuss the results
Comments on the Quality of English LanguageAn extensive English editing revision is suggested
Author Response
Dear Authors, the content of the manuscript is interesting because it gives useful information for diases control in strawberry. Tha experimental design is appropriate, anyway an extensive english editing is required and a more accurate description of methods and results is needed. The major observation are reported below but a complete rewriting of the manuscript could valorize your research
Abstract
Lines 11-30: English editing needs improvement
Response:I have revised the abstract to improved English editing.
Introduction: lines 44-45. “Colletotrichum spp. can be identified by morphological characteristics such as colonies and spores, but it is variable”. What is variable?
Response: The morphological characteristics are easily affected by environmental factors.I have
revised it.
Line 63 -65: “With the continuous use of pyraclostrobin, it is unclear whether the resistance level of strawberry crown rot pathogens has changed.” It is suggested to move this sentence to hypothesis description
Response: Ok, The resistance level of strawberry crown rot pathogens may change with the continuous use of pyraclostrobin
Line 87: please, rephrase “The pathogens were isolated by tissue isolation”
Response: I have rephrased it. The pathogen was isolated from strawberry crown by tissue separation method.
Line 92: “The single-conidium isolates”- Did the authors mean “monosporic cultures”?
Response:Yes.
Line 96: why the authors report this timing “after incubating for 3 days” if mycelium diameter was measured after 7 days? Can the sentence be deleted?
Response: I rewrote the sentence. The isolates stored at a 4℃ were inoculated on a PDA plate incubating for 3 days,and then transferred to a new PDA plate. After incubating for 7 days, the diameter of the colony was measured.
Line 98: the experiment was done in three replication and not “each isolate was repeated 3 times”, I suppose
Response: Ok,I have been modified it.
Line 89-99: rephrase.
Response: Ok,I have been rephrased it make to clear.
Line 101: 50. Write in letters
Response: Ok,changed to “fifty”
Line 109-110: please, rephrase.
Response:Ok,I have been rephrased it
Line 123: “The main pathogens of strawberry crown rot were C. siamense” this sentence have to be moved in results section. The authors could write: “The isolate mostly represented on the basis of molecular identification…..”
Response:I have been modified it.
Line 125: “The main pathogens of strawberry crown rot were C. siamense” this sentence is redundant.
Response: Ok, I've deleted it.
Line 126: What did the authors mean with “edge”?
Response: “edge” means “The outer ring of a colony, the place where the mycelium grows young”
Line 129-130: please, rephrase
Response:Ok, I have been rephrased it to clear.
Section 2.4 and 2.5: I do not understand why the contents of these two section is separated. Did the authors do a pre-screening and then a more accurate evaluation?
Response: Yes, We firstly measured the resistance level, then determined EC50 for further confirmation.
Line 144: please, add more information about cross crossing method
Response: Ok, that was measured horizontally and vertically. I have added it.
Line 162: what did the authors mean with “coated”?
Response: “coated” may not be correct. I have rewritten the sentence.
Line 167-168: please, rephrase
Response: I have been modified this sentence.
Line 178: describe extraction method
Response:I have been added it.
Line 191: describe in materials and methods how the pathogenicity tests were performed
Response: I have been added it.
Line 196-197: give more element to say that the two species have no substantial differences. Briefly explain table 1
Response:table 1 has been described in 3.1.
Figure 2: describe in the text the results in figure 2
Response: figure 2 has been described in 3.1. There was no significant difference in mycelium growth rate,size of spores and appressorium, sporulation between the two Colletotrichum species.
Line 258: this sentence is incorrect.
Response:I have been rephrased it.
Line 262: describe better this analysis in the method and result section
Response:I have been describe this analysis in the method and result section.
Introduction and discussion: as a general suggestion, the introduction is too long and several citation could be moved to discussion to better discuss the result
Response:I have made appropriate deletion to the introduction.
Round 2
Reviewer 1 Report
Comments and Suggestions for Authors
The authors had modified the text according to the comments. I suggest to revise the new version of the manuscript in the following parts:
Lines 63 to 64. Please revise this sentence.
Line 91. Please define ddH2O.
Line 152. Please revise this phrase, the text is not clear.
Line 152. Please revise this phrase, the text is not clear.
Line 171 to 175. Please revise this paragraph.
Line 180. The authors, added new information. Nevertheless, I suggest to include when a spore was considered as germinated (maybe the germ tube length was taken in to account?). Please make it clear.
Line 217. Please use “koch´s postulates” instead of koch´s rule.
Comments on the Quality of English LanguageThe text needs some revision due to English writing issues. I could see that the text can be improved in some parts.
Author Response
Thanks for your suggestions. All are accepted.